# Latent Skill Planning for Exploration and Transfer

**Kevin Xie**[1,*]**, Homanga Bharadhwaj**[1,*]**Danijar Hafner**[1,2]
**Animesh Garg**[1,3]**, Florian Shkurti**[1]
[1]University of Toronto and Vector Institute, [2]Google Brain, [3]Nvidia
ckxie@cs.toronto.edu, homanga@cs.toronto.edu

## Abstract

To quickly solve new tasks in complex environments, intelligent agents need to build up reusable knowledge. For example, a learned world model captures knowledge about the environment that applies to new tasks. Similarly, skills capture general behaviors that can apply to new tasks. In this paper, we investigate how these two approaches can be integrated into a single reinforcement learning agent. Specifically, we leverage the idea of partial amortization for fast adaptation at test time. For this, actions are produced by a policy that is learned over time while the skills it conditions on are chosen using online planning. We demonstrate the benefits of our design decisions across a suite of challenging locomotion tasks and demonstrate improved sample efficiency in single tasks as well as in transfer from one task to another, as compared to competitive baselines. Videos are available at:
https://sites.google.com/view/latent-skill-planning/

## 1 Introduction

Humans can effortlessly compose skills, where skills are a sequence of temporally correlated actions, and quickly adapt skills learned from one task to another. In order to build re-usable knowledge about the environment, Model-based Reinforcement Learning (MBRL) (Wang et al., 2019) provides an intuitive framework which holds the promise of training agents that generalize to different situations, and are sample efficient with respect to number of environment interactions required for training. For temporally composing behaviors, hierarchical reinforcement learning (HRL) (Barto & Mahadevan, 2003) seeks to learn behaviors at different levels of abstraction explicitly.

A simple approach for learning the environment dynamics is to learn a world model either directly in the observation space (Chua et al., 2018; Sharma et al., 2019; Wang & Ba, 2019) or in a latent space (Hafner et al., 2019; 2018). World models summarize an agent's experience in the form of learned transition dynamics, and reward models, which are used to learn either parametric policies by amortizing over the entire training experience (Hafner et al., 2019; Janner et al., 2019), or perform online planning as done in Planet (Hafner et al., 2018), and PETS (Chua et al., 2018). Amortization here refers to learning a parameterized policy, whose parameters are updated using samples during the training phase, and which can then be directly queried at each state to output an action, during evaluation.

Fully online planning methods such as PETS (Chua et al., 2018) only learn the dynamics (and reward) model and rely on an online

Figure 1: Visual illustration of the 2D root position of the quadruped trained with `LSP` on an environment with random obstacles and transferred to this environment with obstacles aligned in a line. The objective is to reach the goal location in red.

---

*Kevin and Homanga contributed equally to this work.

search procedure such as Cross-Entropy Method (CEM; Rubinstein, 1997) on the learned models to determine which action to execute next. Since rollouts from the learned dynamics and reward models are *not* executed in the actual environment during training, these learned models are sometimes also referred to as *imagination* models (Hafner et al., 2018; 2019). Fully amortized methods such as Dreamer (Hafner et al., 2019), train a reactive policy with many rollouts from the imagination model. They then execute the resulting policy in the environment.

The benefit of the amortized method is that it becomes better with experience. Amortized policies are also faster. An action is computed in one forward pass of the reactive policy as opposed to the potentially expensive search procedure used in CEM. Additionally, the performance of the amortized method is more consistent as CEM relies on drawing good samples from a random action distribution. On the other hand, the shortcoming of the amortized policy is generalization. When attempting novel tasks unseen during training, CEM will plan action sequences for the new task, as per the new reward function while a fully amortized method would be stuck with a behaviour optimized for the training tasks. Since it is intractable to perform fully online random shooting based planning in high-dimensional action spaces (Bharadhwaj et al., 2020; Amos & Yarats, 2019), it motivates the question: *can we combine online search with amortized policy learning in a meaningful way to learn useful and transferable skills for MBRL?*

To this end, we propose a partially amortized planning algorithm that temporally composes high-level skills through the Cross-Entropy Method (CEM) (Rubinstein, 1997), and uses these skills to condition a low-level policy that is amortized over the agent's experience. Our world model consists of a learned latent dynamics model, and a learned latent reward model. We have a mutual information (MI) based intrinsic reward objective, in addition to the predicted task rewards that are used to train the low level-policy, while the high level skills are planned through CEM using the learned task rewards. We term our approach `Learning Skills for Planning` (LSP).

The key idea of `LSP` is that the high-level skills are able to abstract out essential information necessary for solving a task, while being agnostic to irrelevant aspects of the environment, such that given a new task in a similar environment, the agent will be able to meaningfully compose the learned skills with very little fine-tuning. In addition, since the skill-space is low dimensional, we can leverage the benefits of online planning in skill space through CEM, without encountering intractability of using CEM for planning directly in the higher dimensional action space and especially for longer time horizons (Figure 1).

In summary, our main contributions are developing a partially amortized planning approach for MBRL, demonstrating that high-level skills can be temporally composed using this scheme to condition low level policies, and experimentally demonstrating the benefit of `LSP` over challenging locomotion tasks that require composing different behaviors to solve the task, and benefit in terms of transfer from one quadruped locomotion task to another, with very little adaptation in the target task.

## 2 BACKGROUND

We discuss learning latent dynamics for MBRL, and mutual information skill discovery, that serve as the basic theoretical tools for our approach.

### 2.1 LEARNING LATENT DYNAMICS AND BEHAVIORS IN IMAGINATION

Latent dynamics models are special cases of world models used in MBRL, that project observations into a latent representation, amenable for planning (Hafner et al., 2019; 2018). This framework is general as it can model both partially observed environments where sensory inputs can be pixel observations, and fully observable environments, where sensory inputs can be proprioceptive state features. The latent dynamics models we consider in this work, consist of four key components, a representation module $p_\theta(s_t|s_{t-1}, a_{t-1}, o_t)$ and an observation module $q_\theta(o_t|s_T)$ that encode observations and actions to continuous vector-valued latent states $s_t$, a latent forward dynamics module $q_\theta(s_t|s_{t-1}, a_{t-1})$ that predicts future latent states given only the past states and actions, and a task reward module $q_\theta(r_t|s_t)$, that predicts the reward from the environment given the current latent state. To learn this model, the agent interacts with the environment and maximizes the following

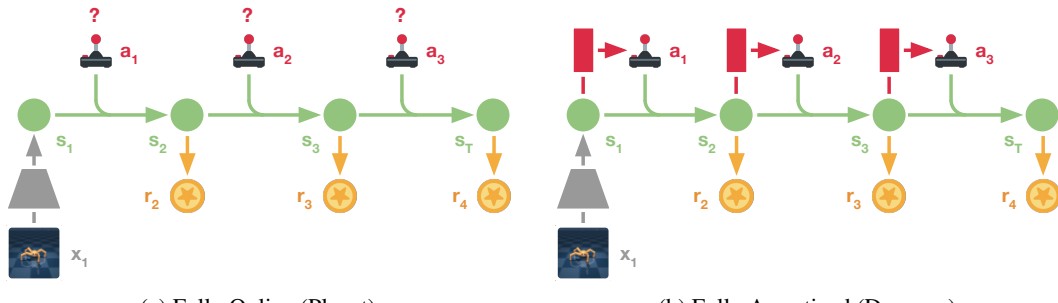

(a) Fully Online (Planet)    (b) Fully Amortized (Dreamer)

Figure 2: **(a)** Online planning where actions are sampled from a distribution with parameters $\theta_i$ **(b)** Fully amortized policy with parameters $\phi$ **(c)** [LSP ] Skills are sampled from a distribution with parameters $\theta_i$, and actions are sampled from a skill-conditioned policy with parameters $\phi$. Here $a$ are actions, $s$ states and $z$ latent plan variables. $\theta_i$ represents parameters of the planning distribution and $\phi$ are the parameters of the policy.

expectation under the dataset of environment interactions $\mathcal{D} = \{(o_t, a_t, r_t)\}$

$$
\mathcal{J} \doteq \mathrm{E}_{\mathcal{D}}\bigg( \sum_t \Big( \mathcal{J}_{\mathrm{O}}^t + \mathcal{J}_{\mathrm{R}}^t + \mathcal{J}_{\mathrm{D}}^t \Big) \bigg) + \mathrm{const} \qquad \mathcal{J}_{\mathrm{O}}^t \doteq \ln q(o_t \mid s_t)
$$
$$
\mathcal{J}_{\mathrm{R}}^t \doteq \ln q(r_t \mid s_t) \qquad \mathcal{J}_{\mathrm{D}}^t \doteq -\beta \, \mathrm{KL}\big( p(s_t \mid s_{t-1}, a_{t-1}, o_t) \,\big\|\, q(s_t \mid s_{t-1}, a_{t-1}) \big).
$$
(1)

For optimizing behavior under this latent dynamics model, the agent rolls out trajectories in imagination and estimates the value $V(\cdot)$ of the imagined trajectories $\{s_\tau, a_\tau, r_\tau\}_{\tau=t}^{t+H}$ through TD($\lambda$) estimates as described by Sutton & Barto (2018); Hafner et al. (2019). The agent can either learn a fully amortized policy $q_\phi(a|s)$ as done in Dreamer, by backpropagating through the learned value network $v_\psi(\cdot)$ or plan online through CEM, for example as in Planet.

## 2.2    MUTUAL INFORMATION SKILL DISCOVERY

Some methods for skill discovery have adopted a probabilistic approach that uses the mutual information between skills and future states as an objective (Sharma et al., 2019). In this approach, skills are represented through a latent variable $z$ upon which a low level policy $\pi(a|s, z)$ is conditioned. Given the current state $s_0$, skills are sampled from some selection distribution $p(z|s_0)$. The skill conditioned policy is executed under the environment dynamics $p_d(s_{t+1}|s_t, a)$ resulting in a series of future states abbreviated $s' := \{s\}$.

Mutual information is defined as:

$$
\mathcal{MI}(z, \{s\}|s_0) = \mathcal{H}(z|s_0) - \mathcal{H}(z|\{s\}, s_0) = \mathcal{H}(\{s\}|s_0) - \mathcal{H}(\{s\}|s_0, z)
$$

It quantifies the reduction in uncertainty about the future states given the skill and vice versa. By maximizing the mutual information with respect to the low level policy, the skills are encouraged to produce discernible future states.

## 3    PARTIAL AMORTIZATION THROUGH HIERARCHY

Our aim is to learn behaviors suitable for solving complex control tasks, and amenable to transfer to different tasks, with minimal fine-tuning. To achieve this, we consider the setting of MBRL, where the agent builds up re-usable knowledge of the environment dynamics. For planning, we adopt a partial amortization strategy, such that some aspects of the behavior are re-used over the entire training experience, while other aspects are learned online. We achieve partial amortization by forming high level latent *plans* and learning a low level *policy* conditioned on the latent plan. The three different forms of amortization in planning are described visually through probabilistic graphical models in Figure 2 and Figure 3.

We first describe the different components of our model, motivate the mutual information based auxiliary objective, and finally discuss the complete algorithm.

**World model.** Our world model is a latent dynamics model consisting of the components described in section 2.

**Low level policy.** The low-level policy $q_\phi(a_t|s_t, z)$ is used to decide which action to execute given the current latent state $s_t$ and the currently active skill $z$. Similar to Dreamer (Hafner et al., 2019), we also train a value model $v_\psi(s_t)$ to estimate the expected rewards the action model achieves from each state $s_t$. We estimate value the same way as in equation 6 of Dreamer, balancing bias and variance. The action model is trained to maximize the estimate of the value, while the value model is trained to fit the estimate of the value that alters as the action model is updated, as done in a typical actor-critic setup (Konda & Tsitsiklis, 2000).

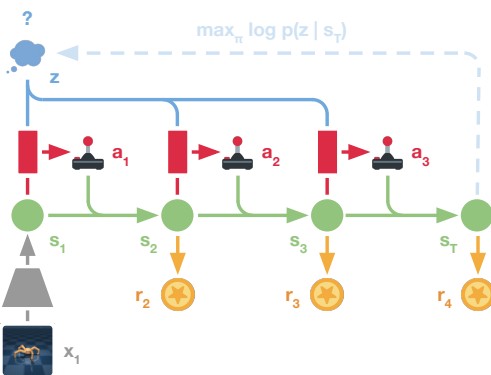

Figure 3: LSP samples kills $z$ from a distribution optimized with CEM online, and samples actions $a$ for every latent state $s$ from skill-conditioned policy. The skill distribution is updated by maximizing mutual information between the skills and the observed latent state distribution. The policy is updated by backpropagating value estimates based on the environment rewards $r$ through the learned latent synamics model.

**High level skills.** In our framework high level skills are continuous random variables that are held for a fixed number $K$ steps. The high-level skills $z$ are sampled from a skill selection distribution $p(z_{1:\lceil H/K \rceil}|\zeta) = \mathcal{N}(\mu, \Sigma)$ which is optimized for task performance through CEM. Here, $H$ denotes the planning horizon. For the sake of notational convenience we denote $z_{1:\lceil H/K \rceil}$ as $z$. Let $^{(j)}$ denote the $j^{\text{th}}$ CEM iteration. We first sample $G$ skills $\{z^{(g)}\}_{g=1}^{G} \sim p(z|\zeta^{(j)})$, execute $G$ parallel imaginary rollouts of horizon $H$ in the learned model with the skill-conditioned policy $q_\phi(a_t|s_t, z^{(g)})$. Instead of evaluating rollouts based only on the sum of rewards, we utilize the value network and compute value estimates $\{V_g\}_{g=1}^{G}$. We sort $\{V_g\}_{g=1}^{G}$, choose the top $M$ values, and use the corresponding skills to update the sampling distribution parameters as $\zeta^{(j+1)} = (\mu^{(j+1)}, \Sigma^{(j+1)})$

$$\mu^{(j+1)} = \text{Mean}(\{z^{(m)}\}_{m=1}^{M}) \quad \Sigma^{(j+1)} = \text{Variance}(\{z^{(m)}\}_{m=1}^{M})$$

### 3.1 OVERALL ALGORITHM

Our overall algorithm consists of the three phases typical in a MBRL pipeline, that are performed iteratively. The complete algorithm is shown in Algorithm 1. The sub-routine for CEM planning that gets called in Algorithm 1 is described in Algorithm 2.

**Model Learning.** We sample a batch of tuples from the dataset of environment interactions $\{(a_t, o_t, r_t)\}_{t=k}^{k+L} \sim \mathcal{D}$, compute the latent states $s_t \sim p_\theta(s_t \mid s_{t-1}, a_{t-1}, o_t)$, and use the resulting data to update the models $p_\theta(s_t \mid s_{t-1}, a_{t-1}, o_t)$, $q_\theta(s_t \mid s_{t-1}, a_{t-1})$, and $q_\theta(r_t \mid s_t)$ through the variational information bottleneck (VIB) (Tishby et al., 2000; Alemi et al., 2016) objective as in equation 13 of Hafner et al. (2019) and as described in section 2.

**Behavior Learning.** Here, we roll out the low-level policy $q_\phi(a_t|s_t, z)$ in the world model and use the state transitions and predicted rewards to optimize the parameters of the policy $\phi$, the skill distribution $\zeta$, the value model $\psi$, and the backward skill predictor $\chi$. The backward skill predictor predicts the skill $z$ given latent rollouts $\{s\}$.

**Environment Interaction.** This step is to collect data in the actual environment for updating the model parameters. Using Model-Predictive Control (MPC), we re-sample high-level skills from the optimized $p_\zeta(z)$ every $K$ steps, and execute the low-level policy $q_\phi(a_t|s_t, z)$, conditioned on the currently active skill $z$. Hence, the latent plan has a lower temporal resolution as compared to the low level policy. This helps us perform temporally abstracted exploration easily in the skill space. We store the (observation, action, reward) tuples in the dataset $\mathcal{D}$.

### 3.2 Mutual Information Skill Objective

Merely conditioning the low level policy $q_\phi(a_t|s_t, z)$ on the skill $z$ is not sufficient as it is prone to ignoring it. Hence, we incorporate maximization of the mutual information (MI) between the latent skills $z$ and the sequence of states $\{s\}$ as an auxiliary objective.

In this paper, we make use of imagination rollouts to estimate the mutual information under the agent's learned dynamics model. We decompose the mutual information in terms of skill uncertainty reduction $\mathcal{MI}(z, \{s\}|s_0) = \mathcal{H}(z|s_0) - \mathcal{H}(z|\{s\}, s_0)$.

**Estimating $\mathcal{MI}(z, \{s\}|s_0)$.** Explicitly writing out the entropy terms, we have

$$\mathcal{MI}(z, \{s\}|s_0) = \mathcal{H}(z|s_0) - \mathcal{H}(z|\{s\}, s_0) = \int p(z, \{s\}, s_0) \log \frac{p(z|\{s\}, s_0)}{p(z|s_0)}$$

In this case we need a tractable approximation to the skill posterior $p(z|s_0, s')$.

$$\mathcal{MI}(z, \{s\}|s_0) = \int p(z, \{s\}, s_0) \left( \log \frac{q(z|\{s\}, s_0)}{p(z|s_0)} + \log \frac{p(z|\{s\}, s_0)}{q(z|\{s\}, s_0)} \right)$$

Here the latter term is a KL divergence and must hence be positive, providing a lower bound for $\mathcal{MI}$.

$$\mathcal{MI}(z, \{s\}|s_0) \geq \int p(z, \{s\}, s_0) \log \frac{q(z|\{s\}, s_0)}{p(z|s_0)}$$

$$= \int p(z, \{s\}, s_0) \log q(z|\{s\}, s_0) - \int p(z|s_0)p(s_0) \log p(z|s_0)$$

$$= \mathbb{E}_{p(z,\{s\},s_0)}[\log q(z|\{s\}, s_0)] + \mathbb{E}_{s_0}[\mathcal{H}[p(z|s_0)]]$$

We parameterize $q(z|\{s\}, s_0)$ with $\chi$, i.e. $q_\chi(z|\{s\}, s_0)$, and call it the backward skill predictor, as it predicts the skill $z$ given latent rollouts $\{s\}$. It is trained through standard supervised learning to maximize the likelihood of imagined rollouts $\mathbb{E}_{p(z,\{s\},s_0)}[\log q(z|\{s\}, s_0)]$. This mutual information objective is only a function of the policy through the first term and hence we use it as the intrinsic reward for the agent $r_i = \log q(z|\{s\}, s_0)$.

The second term $\mathbb{E}_{s_0}[\mathcal{H}[p(z|s_0)]]$ is the entropy of the skill selection distribution. When skills begin to specialize, the CEM distribution will naturally decrease in entropy and so we add Gaussian noise $\epsilon$ to the mean of the CEM-based skill distribution, $\mu \leftarrow \mu + \epsilon$ where $\epsilon \sim \mathcal{N}(0, \mathbb{I}\sigma)$. By doing this we lower bound the entropy of the skill selection distribution.

## 4 Experiments

We perform experimental evaluation over locomotion tasks based on the DeepMind Control Suite framework (Tassa et al., 2018) to understand the following questions:

- Does LSP learn useful skills and compose them appropriately to succeed in individual tasks?
- Does LSP adapt to a target task with different environment reward functions quickly, after being pre-trained on another task?

To answer these, we perform experiments on locomotion tasks, using agents with different dynamics - Quadruped, Walker, Cheetah, and Hopper, and environments where either pixel observations or proprioceptive features are available to the agent. Our experiments consist of evaluation in single tasks, in transfer from one task to another, ablation studies, and visualization of the learned skills.

### 4.1 Setup

**Baselines.** We consider *Dreamer* (Hafner et al., 2019), which is a state of the art model-based RL algorithm with fully amortized policy learning, as the primary baseline, based on its open-source tensorflow2 implementation. We consider a Hierarchical RL (HRL) baseline, HIRO (Nachum et al., 2018) that trains a high level amortized policy (as opposed to high level planning). For consistency, we use the same intrinsic reward for HIRO as our method. We consider two other baselines, named

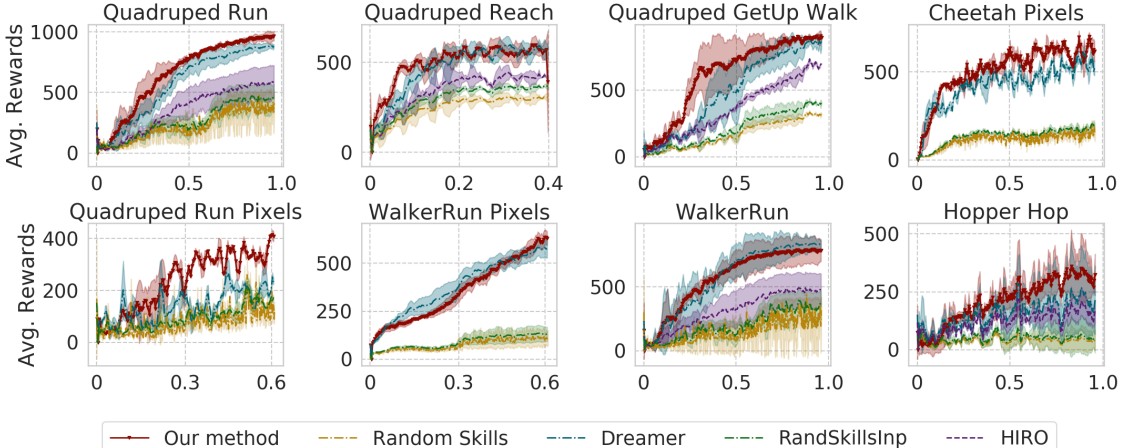

Figure 4: **Single task performance.** x-axis represents number of environment interactions in terms of fraction of 1 Million. Comparison between LSP, and the baselines *Dreamer*, *HIRO*, *RandSkillsInp*, and *Random Skills* on a suite of challenging individual locomotion tasks. Cheetah Pixels, Quadruped Run Pixels, and WalkerRun Pixels are environments with image-based observations while the rest have proprioceptive features (states) as observations to the agent. Higher is better.

*Random Skills* that has a hierarchical structure exactly as LSP but the skills are sampled randomly and there is no planning at the level of skills, and *RandSkillsInit* that is similar to *Random Skills* but does not include the intrinsic rewards (this is essentially equivalent to Dreamer with an additional random skill input) These two baselines are to help understand the utility of the learned skills. For the transfer experiments, we consider an additional baseline, a variant of our method that keeps the low-level policy fixed in the transfer environment. All results are over three random seeds.

**Environments.** We consider challenging locomotion environments from DeepMind Control Suite (Tassa et al., 2018) for evaluation, that require learning walking, running, and hopping gaits which can be achieved by temporally composing skills. In the ***Quadruped GetUp Walk*** task, a quadruped must learn to stand up from a randomly initialized position that is sometimes upside down, and walk on a plane without toppling, while in ***Quadruped Reach***, the quadruped agent must walk in order to reach a particular goal location. In ***Quadruped Run***, the same quadruped agent must run as fast as possible, with higher rewards for faster speed. In the ***Quadruped Obstacle*** environments (Fig. 6a), the quadruped agent must reach a goal while circumventing multiple cylindrical obstacles. In ***Cheetah Run***, and ***Walker Run***, the cheetah and walker agents must run as fast as possible. In ***Hopper Hop***, a one legged hopper must hop in the environment without toppling. It is extremely challenging to maintain stability of this agent.

### 4.2 SOLVING SINGLE LOCOMOTION TASKS.

In Figure 4 we evaluate our approach LSP in comparison to the fully amortized baseline *Dreamer*, and the *Random Skills* baseline on a suite of challenging locomotion tasks. Although the environments have a single task objective, in order to achieve high rewards, the agents need to learn different walking gaits (Quadruped Walk, Walker Walk), running gaits (Quadruped Run, Walker Run), and hopping gaits (Hopper Hop) and compose learned skills appropriately for locomotion.

From the results, it is evident that LSP either outperforms Dreamer or is competitive to it on all the environments. This demonstrates the benefit of the hierarchical skill-based policy learning approach of LSP. In addition, we observe that LSP significantly outperforms the *Random Skills* and *RandSkillsInp* baselines, indicating that learning skills and planning over them is important to succeed in these locomotion tasks. In order to tease out the benefits of hierarchy and partial amortization separately, we consider another hierarchical RL baseline, HIRO (Nachum et al., 2018), which is a state-of-the-art HRL algorithm that learns a high level amortized policy. LSP outperforms HIRO in all the tasks suggesting the utility of temporally composing the learned skills through planning as opposed to amortizing over them with a policy. HIRO has not been shown to work from images

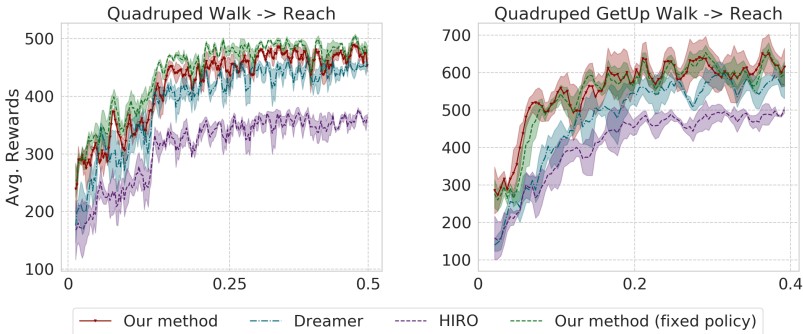

Figure 5: **Transfer**. x-axis represents no. of environment steps in terms of fraction of 1 Million. Comparison between our method (`LSP`), and the baselines *Dreamer*, *HIRO*, and a variant of our method that keeps the policy fixed in the transfer environment. The agents need to transfer to a different locomotion task, after being pretrained on another task. (Higher is better).

in the original paper (Nachum et al., 2018), and so we do not have learning curves for HIRO in the image-based environments, as it cannot scale directly to work with image based observations.

### 4.3 Transfer from one task to another.

In Figure 5 we show results for a quadruped agent that is pre-trained on one task and must transfer to a different task with similar environment dynamics but different task description.

**Quadruped GetUp Walk → Reach Goal.** The quadruped agent is pre-trained on the task of standing up from a randomly initialized position that is sometimes upside down, and walking on a plane without toppling. The transfer task consists of walking to reach a goal, and environment rewards are specified in terms of distance to goal. The agent is randomly initialized and is sometimes initialized upside down, such that it must learn to get upright and then start walking towards the goal. We see that `LSP` can adapt much quickly to the transfer task, achieving a reward of 500 only after 70,000 steps, while Dreamer requires 130,000 steps to achieve the same reward, indicating sample efficient transfer of learned skills.

We observe that the variant of our method `LSP` with a fixed policy in the transfer environment performs as well as or slightly better than `LSP`. This suggests that while transferring from the GetUp Walk to the Reach task, low level control is useful to be directly transferred while planning over high level skills which have changed is essential. As the target task is different, so it requires composition of different skills.

**Quadruped Walk → Reach Goal.** The quadruped agent is randomly initialized, but it is ensured that it is upright at initialization. In this setting, after pre-training, we re-label the value of rewards in the replay buffer of both the baseline *Dreamer*, and `LSP` with the reward function of the target *Quadruped Reach Goal* environment. To do this, we consider each tuple in the replay buffer of imagined trajectories during pre-training, and change the reward labels to the reward value obtained by querying the reward function of the target task at the corresponding state and action of the tuple. From the plot in Figure 5, we see that `LSP` is able to quickly bootstrap learning from the re-labeled Replay Buffer and achieve better target adaptation than the baseline.

From the figure it is evident that for HIRO, the transfer task rewards converge at a much lower value than our method `LSP` and Dreamer, suggesting that the learned skills by an amortized high level policy overfits to the source task, and cannot be efficiently adapted to the target task. Similar to the previous transfer task, we also observe that the variant of our method `LSP` with a fixed policy in the transfer environment performs as well as or slightly better than `LSP`. This provides further evidence that since the underlying dynamics of the Quadruped agent is similar across both the tasks, low level control is useful to be directly transferred while the high level skills need to be adapted through planning.

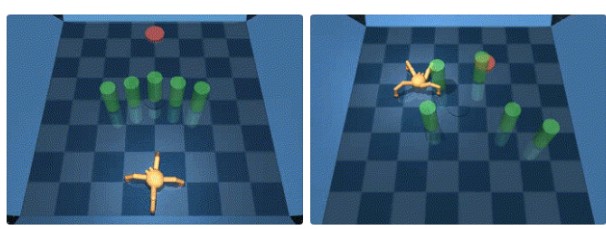 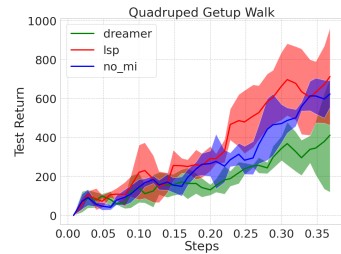

(a) Quadruped obstacle environments        (b) Ablation study

Figure 6: (a) **Quadruped obstacle environments.** Visual illustration of the environment variants we consider. For the environment on the right, the agent must reach the goal while navigating across randomly located cylindrical obstacles while on the left, the agent must walk around a pillar of obstacles to reach the goal. We provide details and results in the appendix. (b) **Ablation study.** x-axis represents no. of environment steps in terms of fraction of 1 Million. Comparison of *dreamer*, `LSP` and the *no_MI* ablation. In *no_MI*, we consider `LSP` but do not use the mutual information maximizing intrinsic reward to train the low level policy, keeping other details exactly the same. Results are on the Quadruped GetUp Walk task. Higher is better.

## 4.4 MUTUAL INFORMATION ABLATION STUDY

In order to better understand the benefit of the mutual information skill objective, we compare performance against a baseline that is equivalent to `LSP` but does not use the intrinsic reward to train the low level policy. We call this ablation baseline that does not have mutual information maximization between skills and states, as *no_MI*. We show the respective reward curves for the Quadruped GetUp Walk task in Figure 6b. Without the mutual information objective, `LSP` learns less quickly but still faster than the baseline *Dreamer*. This emphasizes the necessity of the MI skill objective in section 3.2 and suggests that merely conditioning the low-level policy on the learned skills is still effective to some extent but potentially suffers from the low-level policy learning to ignore them.

## 4.5 VISUALIZATION OF THE LEARNED SKILLS

In Figure 7, we visualize learned skills of `LSP`, while transferring from the Quadruped Walk to the Quadruped Reach task. Each sub-figure (with composited images) corresponds to a different trajectory rolled out from the same initial state. It is evident that the learned skills are reasonably diverse and useful in the transfer task.

## 5 RELATED WORK

**Skill discovery.** Some RL algorithms explicitly try to learn task decompositions in the form of re-usable skills, which are generally formulated as temporally abstracted actions (Sutton et al., 1999). Most recent skill discovery algorithms seek to maximize the mutual information between skills and input observations (Gregor et al., 2016; Florensa et al., 2017), sometimes resulting in an unsupervised diversity maximization objective (Eysenbach et al., 2018; Sharma et al., 2019). DADS (Sharma et al., 2019) is an unsupervised skill discovery algorithm for learning diverse skills with a skill-transition dynamics model, but does not learn a world model for low-level actions and observations, and hence cannot learn through imagined rollouts and instead requiring many environment rollouts with different sampled skills.

**Hierarchical RL.** Hierarchical RL (HRL) (Barto & Mahadevan, 2003) methods decompose a complex task to sub-tasks and solve each task by optimizing a certain objective function. HIRO (Nachum et al., 2018) learns a high-level policy and a low-level policy and computes intrinsic rewards for training the low-level policy through sub-goals specified as part of the state-representation the agent observes. Some other algorithms follow the options framework (Sutton et al., 1999; Bacon et al., 2017), where options correspond to temporal abstractions that need specifying some termination conditions. In practice, it is difficult to learn meaningful termination conditions without additional regularization (Harb et al., 2017). These HRL approaches are inherently specific to the tasks being trained on, and do not necessarily transfer to new domains, even with similar dynamics.

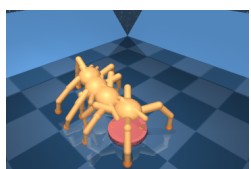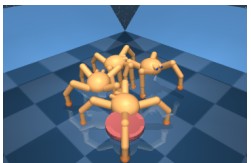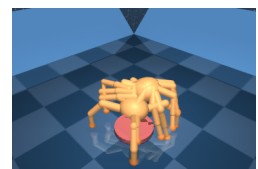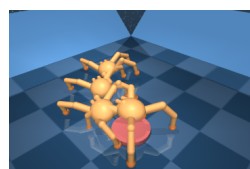

Figure 7: **Qualitative results** visualizing some learned behaviors of `LSP` for the quadruped agent while transferring from the walking to the goal reaching task. Each image depicts the agent performing a different skill for 40 time steps in the environment starting from the same initial state at the origin depicted by the red spot. We see that the learned skills are diverse, correspond to reasonable gaits, and help in performing the task. Video visualizations are in the website https://sites.google.com/view/partial-amortization-hierarchy/home

**Transfer in RL.** Multiple previous works have investigated the problem of transferring policies to different environments. Progressive Networks (Rusu et al., 2016) bootstrap knowledge from previously learned tasks by avoiding catastrophic forgetting of the learned models, (Byravan et al., 2020) perform model-based value estimation for learning an amortized policy and transfer to tasks with different specified reward functions keeping the dynamics the same, while Plan2Explore (Sekar et al., 2020) first learns a global world model without task rewards through a self-supervised objective, and given a user-specified reward function at test time, quickly adapts to it. In contrast to these, several meta RL approaches learn policy parameters that generalize well with little fine-tuning (often in the form of gradient updates) to target environments (Finn et al., 2017; Xu et al., 2018; Wang et al., 2016; Rakelly et al., 2019; Yu et al., 2019).

**Amortization for planning.** Most current MBRL approaches use some version of the 'Cross-Entropy Method' (CEM) or Model-Predictive Path Integral (MPPI) for doing a random population based search of plans given the current model (Wang & Ba, 2019; Hafner et al., 2018; Williams et al., 2016; Sharma et al., 2019). These online non-amortized planning approaches are typically very expensive in high-dimensional action spaces. Although (Wang & Ba, 2019) introduces the idea of performing the CEM search in the parameter space of a distilled policy, it still is very costly and requires a lot of samples for convergence. To mitigate these issues, some recent approaches have combined gradient-descent based planning with CEM (Bharadhwaj et al., 2020; Amos & Yarats, 2019). In contrast, (Janner et al., 2019; Hafner et al., 2019) fully amortize learned policies over the entire training experience, which is fast even for high-dimensional action spaces, but cannot directly transfer to new environments with different dynamics and reward functions. We combined the best of both approaches by using CEM to plan online for high-level skills (of low dimensionality) and amortize the skill conditioned policy for low-level actions (of higher dimensionality).

# 6 DISCUSSION

In this paper, we analyzed the implications of partial amortization with respect to sample efficiency and overall performance on a suite of locomotion and transfer tasks. We specifically focused on the setting where partial amortization is enforced through a hierarchical planning model consisting of a fully amortized low-level policy and a fully online high level skill planner. Through experiments in both state-based and image-based environments we demonstrated the efficacy of our approach in terms of planning for useful skills and executing high-reward achieving policies conditioned on those skills, as evaluated by sample efficiency (measured by number of environment interactions) and asymptotic performance (measured by cumulative rewards and success rate).

One key limitation of our algorithm is that CEM planning is prohibitive in high-dimensional action spaces, and so we cannot have a very high dimensional skill-space for planning with CEM, that might be necessary for learning more expressive/complex skills in real-world robot control tasks. One potential direction of future work is to incorporate amortized learning of skill policies during training, and use CEM for online planning of skills only during inference. Another direction could be to incorporate gradient-descent based planning instead of a random search procedure as CEM, but avoiding local optima in skill planning would be a potential challenge for gradient descent planning.

## ACKNOWLEDGEMENT

We thank Vector Institute Toronto for compute support. We thank Mayank Mittal, Irene Zhang, Alexandra Volokhova, Dylan Turpin, Arthur Allshire, Dhruv Sharma and other members of the UofT CS Robotics group for helpful discussions and feedback on the draft.

## CONTRIBUTIONS

**All the authors were involved in designing the algorithm, shaping the design of experiments, and in writing the paper. Everyone participated in the weekly meetings and brainstorming sessions.**

**Kevin** and **Homanga** led the project by deciding the problem to work on, setting up the coding infrastructure, figuring out details of the algorithm, running experiments, and logging results.

**Danijar** guided the setup of experiments, and helped provide detailed insights when we ran into bottlenecks and helped figure out how to navigate challenges, both with respect to implementation, and algorithm design.

**Animesh** and **Florian** provided valuable insights on what contributions to focus on, and helped us in understanding the limitations of the algorithm at different stages of its development. They also motivated the students to keep going during times of uncertainty and stress induced by the pandemic.

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

# A  APPENDIX

## A.1  ALGORITHM

---

**Algorithm 1:** `Learning Skills for Planning`

---

Initialize dataset $\mathcal{D}$ with $S$ random seed episodes. Initialize neural network parameters $\theta, \phi, \psi$
Define skill duration $K$, plan horizon $H$, CEM population size $G$, MaxCEMiter, skill noise $\epsilon$
**while** *not converged* **do**

    **for** *update step* $c = 1..C$ **do**

        // Model learning
        Draw $B$ data sequences $\{(a_t, o_t, r_t)\}_{t=k}^{k+L} \sim \mathcal{D}$.
        Compute model states $s_t \sim p_\theta(s_t \mid s_{t-1}, a_{t-1}, o_t)$. $\mathcal{S} \leftarrow \mathcal{S} \cup \{s_t\}$
        Update $\theta$ using representation learning.

        // Behavior learning
        $\zeta = (\mu, \Sigma) \leftarrow$ CEM$(\mathcal{S}, \text{MaxCEMiter}, G, H, K, \zeta^{(0)})$; Add noise $\mu = \mu + \epsilon$
        Compute rewards $R = \mathrm{E}\big(q_\theta(r_\tau \mid s_\tau)\big)$ with the optimized CEM distribution $\zeta$
        Compute the corresponding intrinsic rewards $r^i$

        Store the corresponding states into $K-$sized sequences $\{\{s\}_k\}_{k=1}^{\lceil H/K \rceil}$ (for $\chi$)
        Use total rewards $R + r^i$ to form value estimates, and update $\phi, \psi, \chi$

    // Environment interaction
    $o_1 \leftarrow$ `env.reset()`
    **for** *time step* $t = 0..T - 1$ **do**

        // MPC in z space.  Resample skill every K timesteps
        **if** $t\%K == 0$ **then**
            Sample skill $z_{1:\lceil H/K \rceil} \sim p(z_{1:\lceil H/K \rceil}|\zeta)$. Choose the first skill, $z = z_1$
        Compute $s_t \sim p_\theta(s_t \mid s_{t-1}, a_{t-1}, o_t)$ from history, choose $a_t \sim q_\phi(a_t \mid s_t, z)$
        $r_t, o_{t+1} \leftarrow$ `env.step(`$a_t$`)`.
    Add experience to dataset $\mathcal{D} \leftarrow \mathcal{D} \cup \{(o_t, a_t, r_t)_{t=1}^T\}$.

---

## A.2  QUADRUPED OBSTACLE TRANSFER EXPERIMENT

We evaluate the ability of our method to transfer to more complex tasks. Here the source task is to walk forward at a constant speed in a random obstacle environment. The policy is trained in this source task for 500k steps before transferring to the target task which is a pure sparse reward task. The obstacles are arranged in a cove like formation where the straight line to the sparse target leads into a local minima, being stuck against the obstacles. To be able to solve this task, the agent needs to be able to perform long term temporally correlated exploration. We keep all other settings the same but increase the skill length $K$ to 30 time steps and skill horizon $H$ to 120 time steps after transfer in the target task to make the skills be held for longer and let the agent plan further ahead. We see in Figure 10 that the trajectories explored by Dreamer are restricted to be near the initialization and it does not explore beyond the obstacles. In contrast, `LSP` is able to fully explore the environment and reach the sparse goal multiple times. By transferring skills, `LSP` is able to explore at the level of skills and hence produces much more temporally coherent trajectories.

## A.3  SETTINGS AND HYPERPARAMETERS

Our method is based on the tensorflow2 implementation of Dreamer (Hafner et al., 2019) and retains most of the original hyperparameters for policy and model learning. Training is performed more frequently, in particular 1 training update of all modules is done every 5 environment steps. This marginally improves the training performance and is used in the dreamer baseline as well as our method. For feature-based experiments we replace the convolutional encoder and decoder with 2-layer MLP networks with 256 units each and ELU activation. Additionally, the dense decoder network

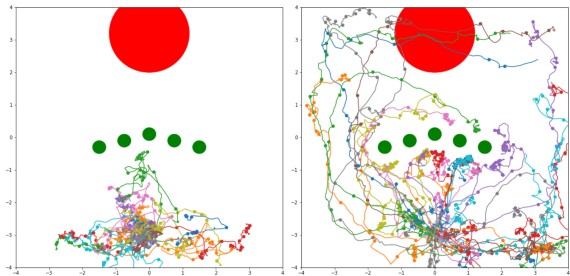

Figure 8: **Quadruped obstacle environment trajectories.** Visual illustration of the 2D root position of the quadruped for the 1st 40k steps after transfer to the cove environment. On the left is Dreamer and on the right is `LSP` with fixed policy. Nodes in the trajectory correspond to beginning state of new skills in the case of `LSP`.

outputs both the mean and log standard deviation of the Gaussian distribution over observations. The standard deviation is softly clamped between 0.1 and 1.5 as described in (Chua et al., 2018).

For `LSP`, skill vectors are 3-dimensional and are held for $K = 10$ steps before being updated. The CEM method has a planning horizon of $H = 10$, goes through $MaxCEMiter = 4$ iterations, proposes $G = 16$ skills and uses the top $M = 4$ proposals to recompute statistics in each iteration. The additional noise $\epsilon$ added to the CEM optimized distribution is Normal(0, 0.1).

The backwards skill predictor shares the same architecture and settings as the feature-based decoder module described above. It is trained with Adam with learning rate of $8e - 5$.

### A.4 CONTROL AS INFERENCE

The control as inference framework (Todorov, 2008; Levine, 2018) provides a heuristic to encourage exploration. It performs inference in a surrogate probabilistic graphical model, where the likelihood of a trajectory being *optimal* (an indicator random variable $\mathcal{O}$) is a hand-designed (monotonically increasing) function of the reward, $\log p(\mathcal{O}|\tau) = f(r)$. The induced surrogate posterior $p(\tau|\mathcal{O})$ places higher probability on higher reward trajectories.

Sampling from $p(\tau|\mathcal{O})$ is in general intractable and so a common solution is to employ variational inference for obtaining an approximation $q_\theta(\tau)$. The objective is to minimize the $\mathbb{KL}$-divergence with respect to the true posterior:

$$\min_\theta \mathbb{KL}(q_\theta(\tau)||p(\tau|\mathcal{O})) = \max_\theta -\mathbb{KL}(q_\theta(\tau)||p(\tau|\mathcal{O})) \tag{2}$$

$$= \max_\theta \mathbb{E}_{q_\theta(\tau)}[\log p(\mathcal{O}|\tau) - \log q_\theta(a)] \tag{3}$$

$$= \max_\theta \mathbb{E}_{q_\theta(\tau)}[f(r)] + \mathcal{H}[q_\theta(a)] \tag{4}$$

Note that this objective boils down to maximizing a function of the reward and an entropy term $\mathcal{H}(\cdot)$.

State of the art model-free reinforcement learning approaches can be interpreted as instances of this framework (Haarnoja et al., 2018). The connection has also been made in model-based RL (Okada & Taniguchi, 2019). Specifically they show how the ubiquitous MPPI/CEM planning methods can be derived from this framework when applying variational inference to simple Gaussian action distributions. Though they show that more sophisticated distributions can be used in the framework as well (in principle), the only other model they investigate in the paper is a mixture Gaussians distribution.

### A.5 HIERARCHICAL INFERENCE

We consider hierarchical action distributions that combine amortizing low level behaviours and online planning by temporally composing these low level behaviours. Specifically we use the following variational distribution:

$$q_{\phi,\theta}(\tau) = \prod_{t=1}^{T} p(s_{t+1}|s_t, a_t) q_\phi(a_t|s_t, z_{k(t)}) \prod_{k=1}^{K} q_{\theta_i}(z_k)$$

Here $z_{1:K}$ are latent variables defining a high level plan that modulates the behaviour of the low level policy $q_\phi(a_t|s_t, z_{k(t)})$. $k(t)$ is an assignment defining which of the $K$ $z$'s will act at time $t$. For all our experiments, we chose a fixed size window assignment such that $k(t) = \lfloor t(K/T) \rfloor$.

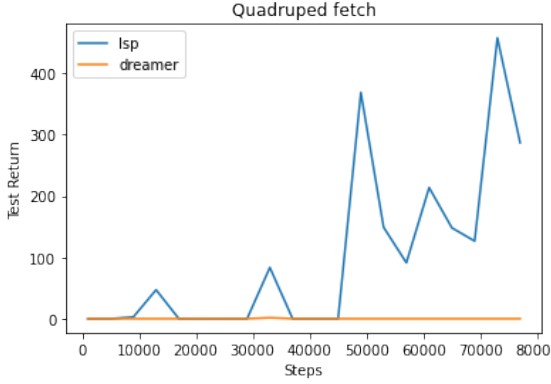

Figure 9: **Quadruped obstacle environment transfer learning curves.** We plot total rewards against number of transfer training time steps on the sparse cove environment. Note that after 70k steps (70 episodes), the LSP agent is able reliably rech the target every time, whereas dreamer does not even come close to the target, stunted by the sparse reward.

Here $p(\mathcal{T})$ is the task distribution over which we wish to amortize. For example using $p(s_0)$ would amortize over plans starting from different initial states. Note that the minimization over $\phi$ is outside the expectation which means that it is amortized, whereas the minimization over $\theta$ is inside such that it is inferred online.

$$\min_{\phi} \mathbb{E}_{p(\mathcal{T})}[\min_{\theta} KL(q_\theta(\tau)||p(\tau|\mathcal{O}))] \tag{5}$$

$$= \max_{\phi} \mathbb{E}_{p(\mathcal{T})}[\max_{\theta} \mathbb{E}_{q_{\phi,\theta}(\tau)}[\log p(\mathcal{O}|\tau)p_0(\mathbf{z}) - \log q_\phi(\mathbf{a}|\mathbf{s},\mathbf{z})q_{\theta_i}(\mathbf{z})]] \tag{6}$$

This formulation also directly relates to some meta-learning approaches that can be interpreted as hierarchical Bayes (Grant et al., 2018). There are different options for the inner and outer optimization in this type of meta-optimization.

### A.6 NOTE ON MUTUAL INFORMATION

In the paper we used the reverse skill predictor approach to estimate the mutual information objective. The alternate decomposition in terms of reduction in future state uncertainty is given by:

$$\mathcal{H}(\{s\}|s_0) - \mathcal{H}(\{s\}|z, s_0) = \int p(z, \{s\}, s_0) \log \frac{p(\{s\}|z, s_0)}{p(\{s\}|s_0)}$$

In this case tractable approximations of the the skill dynamics $p(s'|z, s_0)$ and marginal dynamics $p(s'|s_0)$ are required.

In theory, these can be formed exactly as compositions of the policy, skill selection distribution and dynamics model:

$$\log p(\{s\}|z, s_0) = \log \prod_t p(s_{t+1}|s_t, a)\pi(a|s_t, z)$$

$$\log p(\{s\}|s_0) = \log \mathbb{E}_{z'|s_0}[p(\{s\}|z', s_0)]$$

The difficulty in using this formulation in our case is due to the marginal future state distribution $p(\{s\}|s_0)$. To form the marginal we rely on a finite number $K$ monte carlo samples which is biased and our lower bound on mutual information will be reduced by $\log K$. This means that for every given initial state we wish to train on, we must also sample a sufficiently large number of skills from the skill selection distribution to form an acceptable marginal approximation, greatly increasing the computation cost compared to using the reverse skill predictor.

### A.7 MUTUAL INFORMATION SKLL PREDICTOR NOISE

Although mutual information is a popular objective for skill discovery, it is important to understand its limitations. When the underlying dynamics of the system are deterministic, a powerful world model may be able to make future predictions with high certainty. Consider if the agent is then able to communicate the skill identity to the skill predictor through a smaller subspace of the state

space. For example, imagine that in a quadruped walking task the height of a specific end effector at a certain future frame can be very confidently predicted by the dynamics model. Whether to lift the end effector up 9cm or 10cm may be largely negligible in terms of the dynamics of the motion, but the policy may elect to map the skill variable to the heights of a specific end effector at a certain future frame. When this occurs, diverse skills can become quite indistinguishable visually yet be very distinguishable by the skill predictor. In this failure case, the mutual information may be minimized without truly increasing the diversity of the motion. We use a simple idea to combat this by artificially adding noise to the input of the skill predictor. The higher the skill predictor input noise is, the harder it is to distinguish similar trajectories which forces the skills to produce more distinct outcomes.

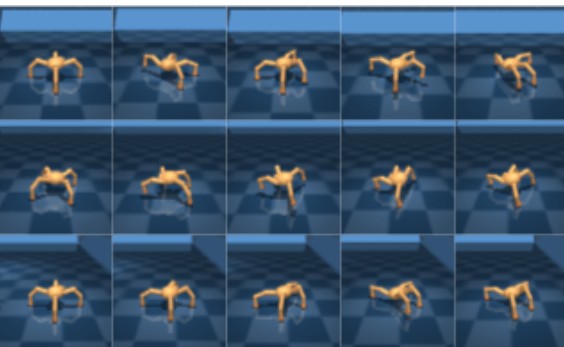

Figure 10: **Example of collapsed skill policy.** Here we show skill sequences sampled from a policy where input noise is not added to the skill predictor and subsequently the skill policy seems to perform the same motion for different skills (turning clockwise), despite inclusion of the mutual information objective. Each row is a different latent variable.

---

**Algorithm 2:** CEM subroutine for Algorithm 1

---

**Function** CEM ($\mathcal{S}$, *MaxCEMiter, G, H, K*, $\zeta^{(0)}$) **:**

    Sample $G$ initial skills $\{z^{(g)}_{1:\lceil H/K \rceil}\}^{G}_{g=1}$ from prior $p(z|\zeta^{(0)})$

    **for** *j in range(MaxCEMiter)* **do**

        **for** *g in range(G)* **do**

            **for** *each $s_t$* **do**

                $\tau = t$ // Imagine trajectories $\{(s_\tau, a_\tau)\}^{t+H}_{\tau=t}$ from each $s_t$.

                **while** $\tau < H$, $\tau ++$ **do**

                    **if** $(\tau - t)\%K == 0$ **then**

                        Sample skill $z_{1:\lceil H/K \rceil} \sim p(z_{1:\lceil H/K \rceil}|\zeta^{(j)})$

                    Sample action $a_\tau \sim q_\phi(a_\tau \mid s_\tau, z_1)$ // Choose $z_1$ via MPC

                    Observe new state $s_{\tau+1}$ and compute new intrinsic reward $r_i$

            Compute value estimates $V_g$ from rewards of each imagined rollout

        **Sort** $\{V_g\}^{G}_{g=1}$ and use the top M sequences to update $\zeta^{(j)} = (\mu^{(j)}, \Sigma^{(j)})$

        Sample $G$ skills $\{z^{(g)}\}^{G}_{g=1}$ from $p(z|\zeta^{(j)})$

    **return** $\zeta^{(\text{MaxCEMiter})}$

---

