# OpenReview forum: "Latent Skill Planning for Exploration and Transfer"
_ICLR.cc/2021/Conference — ICLR 2021 Poster_

### Official Review · AnonReviewer4 · 2020-10-26
**Hierarchical model-base RL for skill transfer -- interesting idea but not sufficiently convincing**

**Rating:** 6
**Confidence:** 3

**Review:**

Summary:
This paper presents a model-based RL approach that i) learns a high-level plan generator that produces a sequence of latent continuous variables as skills for a task through CEM, and ii) a low-level policy conditioned on the skills amortized policy training. The approach is shown to be more sample efficient than a recent model-base RL approach (Dreamer), and is able to learn to solve a new (but related) in simple walking tasks relatively faster as well.

Strengths:
+ The main idea (planning over low-level dimension and learning amortized low-level policy over high-dimension) is intriguing and makes a lot of sense, especially compared to alternative learning schemes such as purely CEM or purely amortized policy training.

+ The approach is well motivated and is easy to understand.

+ The results in simple tasks show promising results over existing approaches.

+ I find the background and the discussion of the connection between existing model-based approaches and the proposed method very helpful.

Weakness:
- The high-level skill planner / generator p(z |\cdot) does not seem to rely on the physical states, which is different from typical HRL approaches where the option/subgoal at each step depends on the latest state, e.g., p(z^t | s^t). This seems to suggest that the plan of skills for a task is somewhat fixed regardless of the initial or immediate states. This may work when the environment is simple and does not vary a lot from episode to episode; but I can not see how this formulation can tackle more complex scenarios. Moreover, a fixed hyper-parameter 'K' seems to require additional tuning for new tasks and environments, which may make the approach less 'generalizable.'

- On a related note, it would be good to see more discussion on the advantages of the proposed skill generator/planner over the prior work on HRL. The related work section briefly mentions some prior work, but there is no detailed theoretical and/or empirical comparison. As of now, it is not clear to me how this is better than conventional HRL methods (e.g., learning a high-level policy).

- The main objective of the approach seems to be improving the transferability of skills learned by RL as stated in the main text and in the title. However, the transfer setting is quite simple (from 'learning to walking' to 'learning to walk to a goal location'). I would be more interested in seeing more difficult settings such as walking in new and complex environments (e.g., with obstacles). It is also not clear to me what exactly is transferred here. Is it the low-level policy? Is it the high-level skill plan? Based on my understanding, the authors may be aiming at the former. If so, a transfer setting where the low-level policies are fixed and only the high-level plans are updated / fine-tuned would be far more convincing. After all, I think that is what people would normally expect when you are claiming skill transfer, isn't? In fact, to my knowledge, some HRL approaches have attempted to reuse previously learned & fixed policies (e.g., [1,2]). I'm not saying that these approaches are better, but since the focus is skill transfer, I think a proper discussion/comparison would be critical.

For rebuttal, please address my concerns listed above. I think the paper presents an interesting and promising idea, but I am not fully convinced that the approach has the advantages as claimed in the paper.

References:

[1] Tessler et al., A Deep Hierarchical Approach to Lifelong Learning in Minecraft, AAAI 2017.

[2] Shu et al., Hierarchical and Interpretable Skill Acquisition in Multi-task Reinforcement Learning, ICLR 2018.

---

> ### Author Response · Authors · 2020-11-18
> **Author response: We have added results for a HRL baseline, and a baseline for transfer with fixed policy**
>
> Thank you for the detailed review and comments about our paper. The main points raised in the review are clarification about the skill planner, addition of a HRL baseline, and addition of a baseline for transfer that keeps the policy fixed. We have clarified the first point below, added a strong model-free HRL baseline HIRO [1] (in Figures 3 and 4) and added a baseline that keeps the low-level policy fixed during transfer (Figure 4). We have updated the related works section with discussions about HRL approaches.
>
>
> We would request the reviewer to revisit the paper in light of these revisions and clarifications, and let us know if any further reservations remain. We have highlighted major modifications, wherever possible in blue.
>
> In the points below, we first paraphrase texts from the review, and provide our responses in bold.
>
>
> The high-level skill planner / generator $p(z |\cdot)$ does not seem to rely on the physical states, which is different from typical HRL approaches where the option/subgoal at each step depends on the latest state, e.g., $p(z^t | s^t)$.  **There might be a point of confusion here. We would like to point out that the MPC procedure makes use of the latest state in generating the parameters $\zeta$ of the parametrized distribution $p(z |\zeta)$ over skills, specifically the current state is where the rollouts of MPC start from. This is different from the induced conditional probability as the MPC procedure itself is also stochastic.**
>
> Moreover, a fixed hyper-parameter 'K' seems to require additional tuning for new tasks and environments, which may make the approach less 'generalizable.' **In all our reported results we have used the same fixed K and did not tune this. We agree that it would be ideal to have an automatic way of tuning the value of 'K' during training, but we could not devise a simple way of doing it without adding a large complexity overhead to the model (for example, by meta-learning K).**
>
> On a related note, it would be good to see more discussion on the advantages of the proposed skill generator/planner over the prior work on HRL. The related work section briefly mentions some prior work, but there is no detailed theoretical and/or empirical comparison. As of now, it is not clear to me how this is better than conventional HRL methods (e.g., learning a high-level policy). **We agree with this completely. The main intuition for us to study a model-based setting is because we can clearly isolate dynamics learning and planning using the learned dynamics - so we can analyze transfer to settings with similar dynamics but different tasks. We have now added a strong model-free HRL baseline, HIRO. (in Figures 3 and 4). HIRO does exactly what you mentioned - learning a high level policy. For a fair comparison, we used the same intrinsic reward for HIRO as our approach (in the original HIRO paper, the intrinsic rewards correspond to intermediate goals, but that would require additional information from the environment to be incorporated for this baseline, and the comparison to our method would not be fair.)**
>
> The main objective of the approach seems to be improving the transferability of skills learned by RL as stated in the main text and in the title. However, the transfer setting is quite simple (from 'learning to walking' to 'learning to walk to a goal location'). I would be more interested in seeing more difficult settings such as walking in new and complex environments (e.g., with obstacles). **Our method focuses on learning environment dynamics and low level skills that can be used in the environment for planning. As such we did not consider transfer cases where the low level dynamics of the target task differs from the source task as would be the case for ``walking in new and complex environments (e.g., with obstacles)".**
>
> It is also not clear to me what exactly is transferred here. Is it the low-level policy? Is it the high-level skill plan? **It is the low level policy, and how the policy is conditioned by high level skills. Additionally, the learned dynamics model is also transferred.**
>
> If so, a transfer setting where the low-level policies are fixed and only the high-level plans are updated / fine-tuned would be far more convincing. **Thank you for suggesting this. We have now added this comparison in the transfer setting in Figure 4. This performs slightly better than the original version of our method LSP used in transfer and is consistent with our previous results.**
>
> [1] Nachum, O., Gu, S.S., Lee, H. and Levine, S., 2018. Data-efficient hierarchical reinforcement learning. In Advances in Neural Information Processing Systems (pp. 3303-3313).

---

> > ### Comment · AnonReviewer4 · 2020-11-22
> > **Thanks for the clarification and new results**
> >
> > Thanks for the clarification and the new experimental results! In particular, it is good (and surprising) to see that with fixed low-level policies, LSP can achieve a sightly better performance than the original version. I have to admit that this is a bit counter intuitive to me. This is not a criticism, but could you explain why a harder transfer setting where you don't refine the low-level policies can give you slightly better results? Also, I can't find the curves for HIRO in Figure 3 for the cheetah pixels, quadruped run pixels, and walkerrun pixels tasks. Is that a mistake or is there no evaluation of HIRO in environments with image-based observations? Lastly, I would encourage the authors to provide more discussions on all the new experimental results in the main text (currently I can only see the new curves in the figures).

---

> > > ### Author Response · Authors · 2020-11-23
> > > **Thank you for the revised assessment of our updated paper. We have now added details about the new experimental results in the main text of the paper.**
> > >
> > > Thank you for acknowledging the new results we have added in the paper. We have updated the paper with discussions about the new results in the main text of Section 4. We would be grateful if you kindly let us know if there is anything else we should clarify for a revised positive assessment of our paper and the rating score.
> > >
> > > We have provided the clarifications based on your questions, pointwise below.
> > >
> > > This is not a criticism, but could you explain why a harder transfer setting where you don't refine the low-level policies can give you slightly better results? **We believe that since the dynamics of the Quadruped agent is the same across the source and target environments, while the task description has changed, the low level policy can be adapted as it is without tuning in the target environment. Doing this yields slightly better or as good results as with updating the low level policy in the target environment. While, we still need to re-plan and update the skill distribution in the target environment, because the high-level skills which must be composed to solve the new task have changed from the source task. We have added details explaining this in Section 4.4 of the revised paper.**
> > >
> > >
> > > Is that a mistake or is there no evaluation of HIRO in environments with image-based observations? **We would like to kindly note that HIRO has not been shown to work from images in the original paper [1], and so we do not have learning curves for HIRO in the image-based environments, as it cannot scale directly to work with image based observations.**
> > >
> > > Lastly, I would encourage the authors to provide more discussions on all the new experimental results in the main text **Thank you for pointing this out. We have now updated Section 4 to include additional interpretations and intuitions about the new results in the main text of the paper. Kindly let us know if there is anything else we should clarify.**
> > >
> > > We would be happy to provide additional clarifications, and address any other concerns that the reviewer might have for this paper.
> > >
> > > [1] Nachum, O., Gu, S.S., Lee, H. and Levine, S., 2018. Data-efficient hierarchical reinforcement learning. In Advances in Neural Information Processing Systems (pp. 3303-3313).

---

> > > > ### Comment · AnonReviewer4 · 2020-11-23
> > > > **Thanks for the follow up responses**
> > > >
> > > > Thanks for the clarification. I am happy with the responses and will increase my rating to 6 accordingly.
> > > >
> > > > Minor comments:
> > > > 1. Please explain why HIRO was not evaluated in the environments with raw pixels as inputs in the paper.
> > > > 2. In Section 4.4, the last paragraph:
> > > > a. From the Figure->  From the figure
> > > > b.  converge at a much lower value that our method LSP and Dreamer: that -> than

---

> > > > > ### Author Response · Authors · 2020-11-23
> > > > > **Thank you!**
> > > > >
> > > > > Thank you for the revised assessment of our paper. We have now addressed the final comments in the updated paper.

---

> ### Author Response · Authors · 2020-11-20
> **Discussion**
>
> Kindly let us know if our response below addressed your concerns. We will be happy to answer if there are additional issues/questions.

---

### Official Review · AnonReviewer3 · 2020-10-28
**Interesting fusion of skill discovery and latent space planning with shortcomings in presentation in experimental section.**

**Rating:** 5
**Confidence:** 4

**Review:**

The authors propose an extension of the Dreamer method (Hafner et al., 2019) to incorporate high-level planning via the cross-entropy method and temporally extended skill policies. In theory, this would allow the planning method to operate in an abstract, high-level space and delegate low-level action selection to policies trained on simulated rollouts. If I understand the paper correctly, the result of the LSP learning algorithm is the skill policy and a (fixed?) distribution over a skill variable; no planning is performed at test time. This only became clear when reading the section on further work, since the abstract states "For this, actions are produced by a policy that is learned over time while the skills it conditions on are chosen using online planning.". If there is indeed no planning during evaluation I would propose that "online planning" is not an apt description of the architecture. I general however, I like how the presented algorithm mixes MI-based skill discovery (which is typically unsupervised) with explicit supervision via long-horizon planning and would regard this a useful contribution to the current literature on hierarchical RL.

My rating is mainly influenced by the experimental section, as well as several minor issues with the overall presentation. The experiments show that the method is at least as good, and sometimes better, than Dreamer when training from scratch, and universally (although marginally) better when adapting to new tasks.  This is a good result, but some relevant details on the tasks and plots are missing, which make the achievements hard to interpret (see below). The difference to Dreamer is most pronounced on the "Quadruped Run Pixels" task, but as mentioned before it's not easy to judge the results; for example, in Figure 8 in the Dreamer paper they report scores of ~750 after 2e6 environment steps. It would also be helpful to provide reference numbers for a standard model-free algorithm on those tasks. The "Random Skills" baseline performs very poorly; does it still include the intrinsic rewards to ensure diversity of skills? What would happen if you would drop them so that you basically have Dreamer with an additional random skill input (cf. the "Switching Ensemble" baseline from [Why Does Hierarchy (Sometimes) Work So Well in Reinforcement Learning?](https://arxiv.org/abs/1909.10618))? Finally, the experiments don't give much insight into the nature of skills that have been discovered. In particular, what do the resulting skill distributions look like?

Several aspects of the experimental section are unclear to me and hinder the interpretation of the results:
- Several crucial details on the environments are missing, such as the reward functions. For "Quadruped Reach", is the goal constant or randomly selected for each episode?
- In Figure 3, what does the x-Axis represent?
- In Figure 3, tasks like "Quadruped Run", "WalkerRun Pixels" and "Quadruped Run Pixels" don't seem to be trained to convergence.
- There is a significant drop in performance at x=0.4 for "Quadruped Reach" for some seeds.
- In 4.4., the text for "Quadruped GetUp Walk -> Reach Goal" does not line up with the descriptions in 4.1. as it says the pretraining task would be to run as fast as possible rather than getting up and walking.
- In the same paragraph: "We see that LSP can adapt much quickly to the transfer task, achieving a reward of 500 only after 70, 000 steps, while Dreamer requires 130, 000 steps to achieve the same reward". From Figure 4 this appears to be the result for "Quadruped Walk -> Reach Goal", and in addition it's also worth noting that Dreamer catches up pretty soon afterwards (even though LSP maintains higher performance throughout training).
- For the transfer experiments, it would be useful to include the learning curves from Figure 3 so that it is apparent whether it is generally beneficial to adapt a pretrained model as opposed to learning from scratch.

A few more notes on the overall write-up:
- The site with videos linked to in the abstract is empty
- In the introduction, it's not immediately clear what "amortized" means even though it is mentioned several times. My interpretation is that it means that you use a learned model to train a policy rather than using the model during evaluation. The term could be described more explicitly; in particular I was slightly confused by the notion of "amortizing over the entire training experience".
- At the end of Section 5, regarding Dreamer: "[...], which is fast even for high-dimensional action spaces, but cannot directly transfer to new environments with different dynamics and reward functions." This is in contradiction to the transfer experiments in 4.4. which show a good amount of transfer performance?
- Figure 2b) and c): $\phi$ in text vs. $\Phi$ in figure.
- In section 3 the reference should be to Figure 2 rather than 2b)?
- In section 3, "Low level policy", it would be good to cite the Dreamer paper once more.
- In 3.1, "Behavior Learning.", the backwards skill predictor is mentioned but not introduced yet. Refer to section 3.2 for a definition?
- Section 3.2.: it would be good to relate the MI objective to previous methods. The resulting objective looks quite similar to [Variational Intrinsic Control](https://arxiv.org/abs/1611.07507).
- Section 5: DADS is mentioned, but it might be also be worth noting that they performed experiments with online planning on top of the discovered skills.

---

> ### Author Response · Authors · 2020-11-18
> **Author response: We have added results for a new Random Skills and a Hierarchical RL baseline**
>
> Thank you for the detailed review and comments about our paper. The major review comments were about addition of new baselines, and issues with the clarity of writing/presentation in some places. We have now added two baselines - Dreamer with an additional random skill input (as mentioned by the reviewer), and a strong model-free Hierarchical RL baseline, HIRO [1]. In addition we have addressed the concerns with presentation and provided clarifications.
>
> We would request the reviewer to revisit the paper in light of these revisions and clarifications, and let us know if any further reservations remain. We have highlighted major modifications, wherever possible in blue.
>
> In the points below, we first paraphrase texts from the review, and provide our responses in bold.
>
>
> If I understand the paper correctly, the result of the LSP learning algorithm is the skill policy and a (fixed?) distribution over a skill variable; no planning is performed at test time. **We would like to kindly clarify a misunderstanding here and point out that at test time we in fact do planning with MPC and the skill distribution is not fixed.**
>
> The "Random Skills" baseline performs very poorly; does it still include the intrinsic rewards to ensure diversity of skills? What would happen if you would drop them so that you basically have Dreamer with an additional random skill input.  **We have now added this baseline named "RandSkillsInp" in Figure 3 of the paper. This performs slightly better than the previous Random Skills baseline (in yellow), but still the performance of this is not good. We believe that this is a strong indication of the utility of learned skills.**
>
> It would also be helpful to provide reference numbers for a standard model-free algorithm on those tasks. **We have also added results for comparison with a strong model-free Hierarchical RL baseline, HIRO based on the authors' code that learns a high level policy and a low level policy, in Figures 3 and 4 of the revised paper.**
>
> Several crucial details on the environments are missing, such as the reward functions. For "Quadruped Reach", is the goal constant or randomly selected for each episode? **For the Quadruped Run, Quadruped Walk, Cheetah Run, Walker Run, Hopper Hop environments we used the standard reward functions from DM Control https://github.com/deepmind/dm_control For the Quadruped Reach task, the goal is randomly selected at the start of each episode, and there is a dense reward signal (as a function of the distance to goal) for the agent to reach the goal**
>
> In Figure 3, what does the x-Axis represent? **We have mentioned in the first line of the caption that x-axis represents number of environment interactions in terms of fraction of 1 Million. We did not include it in the Figure itself due to space constraint in the figure**
>
> There is a significant drop in performance at x=0.4 for "Quadruped Reach" for some seeds. **We apologize for the confusion. The drop at the very last datapoint showed up due to smoothing issues during plotting, and is not an actual drop in the performance of any of the seeds**
>
> In 4.4., the text for "Quadruped GetUp Walk -> Reach Goal" does not line up with the descriptions in 4.1 **Thank you for pointing this out. We have fixed this line now.**
>
> The site with videos linked to in the abstract is empty. **Sorry about this. We have fixed it now.**
>
> In the introduction, it's not immediately clear what "amortized" means even though it is mentioned several times. **We have clarified this now. Amortization here refers to learning a parameterized policy, whose parameters are updated using samples during the training phase, and which can then be directly queried at each state to output an action, during evaluation.**
>
> Figure 2b) and c). **We have made the notations in the figure and the caption consistent now**
>
> In section 3 the reference should be to Figure 2 rather than 2b)? **Yes, we have fixed this.**
>
> In section 3, "Low level policy", it would be good to cite the Dreamer paper once more. **Thanks. We have done this now.**
>
> In 3.1, "Behavior Learning.", the backwards skill predictor is mentioned but not introduced yet. Refer to section 3.2 for a definition?   **We have added the definition in section 3.1 as well now.**
>
> [1] Nachum, O., Gu, S.S., Lee, H. and Levine, S., 2018. Data-efficient hierarchical reinforcement learning. In Advances in Neural Information Processing Systems (pp. 3303-3313).

---

> > ### Comment · AnonReviewer3 · 2020-11-24
> > **Thanks for the clarification and updates**
> >
> > Thank you very much for the clarification and updates provided. Considering my initial misunderstanding of a fixed skill distribution at test time, I'll raise my score to 5.
> >
> > Regarding the model-free baseline, I actually wanted to see a non-hierarchical model-free baseline (sorry for not being 100% clear on this). For example, similar to Dreamer, it would be good to add a "top-line" of the result that D4PG achieves at convergence. These numbers are available in the dm_control paper. This way, it would be clear that the proposed algorithm learns faster but might not be able to reach the final performance of a non-hierarchical method (for example, https://arxiv.org/abs/1801.00690v1 reports a mean score of 560.4 for "Hopper Hop" after 1e8 steps, while your algorithm achieves ~300). However, I definitely also appreciate that you added the comparison to HIRO.
> >
> > It would be great if you could fix the plotting error for Quadruped Reach. I'm also still not clear regarding the discrepancy of the Dreamer learning curve in Figure 3 for "Quadruped Run Pixels" compared to the Figure 8 in the Dreamer paper. After 1e6 steps, the Dreamer paper reports a score of about 500, and it's 750 after 2e6 steps. In your figure, it achieves ~200 after 1e6 steps. Similarly, (assume these are the same tasks), your numbers for Dreamer for "Cheetah Pixels" just makes it over 500, while in the Dreamer paper 750 is obtained in less than 0.5e6 steps for "Cheetah Run".

---

> > > ### Author Response · Authors · 2020-11-25
> > > **Thank you for your revised assessment and comments about the updated results.**
> > >
> > > We thank the reviewer for their reply and revised assessment of our paper.
> > >
> > > Thank you for clarifying what you meant by the model-free baselines. We agree that it will be useful to have this in the plots for comparison of asymptotic performance. We will include these, and also update the Quadruped Reach figure after resolving the plotting error. We will do this in the final revised version as there is very less time now before the author response closes. Thank you for your understanding.
> > >
> > > Regarding the "Quadruped Run Pixels" Dreamer baseline, there might be a slight misunderstanding. Our plot goes only upto 0.6e6 (not 1e6). We will run these for a little longer upto 1e6 and revise the plots, but in the paper we focused the comparison in that region because we expect our method to be sample efficient early on during training.
> > >
> > > Regarding "Cheetah Pixels" the slight discrepancy with respect to the results reported in the Dreamer paper may be because we ran the Dreamer baseline code ourselves and reported the results we got (instead of directly adopting the results from the paper). We thank you for pointing this out. We will now run this baseline for many more seeds and report the resulting performance.

---

> ### Author Response · Authors · 2020-11-20
> **Discussion**
>
> Kindly let us know if our response below addressed your concerns. We will be happy to answer if there are additional issues/questions.

---

### Official Review · AnonReviewer2 · 2020-10-28
**novel approach of transferable hierarchical planning**

**Rating:** 7
**Confidence:** 4

**Review:**

The paper deals with learning reusable hierarchical skills in sequential decisions.
The authors introduce a partially amortized model based on four parametric functions (the representation module, observation module, latent forward dynamic, and task reward module) and an additive loss function composing these modules.
The overall algorithm consists of learning the four-modules jointly and to plan using MPC in the latent skill space.
The overall paper is well written and justified.
The experiments are done in the locomotion domain and show better or comparables results wrt to DREAM.
DADS is mentioned in the SoA section, it would have been interested to evaluate against it.
Furthermore, despite its computational complexity as mentioned in the discussion section, I would have been interested to have a comparison or discussion wrt [1].
Overall the paper is original and sufficiently empirically motivated for acceptance.
I would request the code to be open-source for better reproducibility.

Refs:
[1] Emergent Real-World Robotic Skills via Unsupervised Off-Policy Reinforcement Learning, Levine and al, 2020

---

> ### Author Response · Authors · 2020-11-18
> **Author response: Comparison**
>
> Thank you for the detailed review and comments about our paper. We will definitely open-source the code for our paper. We have answered your comment below. Our response is in bold.
>
> The experiments are done in the locomotion domain and show better or comparable results wrt to DREAM. DADS is mentioned in the SoA section, it would have been interested to evaluate against it. Furthermore, despite its computational complexity as mentioned in the discussion section, I would have been interested to have a comparison or discussion wrt [1]. **We would like to kindly point out that DADS and the follow-up to DADS [1] requires privileged information in the form of the ground truth test-time reward function, and simulator states. DADS cannot handle image-based observations, so it is not possible to have a direct comparison with it empirically.**
>
> [1] Emergent Real-World Robotic Skills via Unsupervised Off-Policy Reinforcement Learning, Levine and al, 2020

---

### Official Review · AnonReviewer1 · 2020-10-29
**SKILL TRANSFER VIA PARTIALLY AMORTIZED HIERARCHICAL PLANNING**

**Rating:** 6
**Confidence:** 3

**Review:**

The paper proposes combining model-based RL with high-level skill learning and composition through hierarchical RL, into a single reinforcement learning framework. More specifically, the proposed approach leverages planning and composing skills in the low-dimensional, high-level representation, and learn low-level skills conditioned on the high-level skills. Only the low-level policies are executed in the environment to generate experiences. A mutual information objective is used to learn low-level policies conditioned on high-level skills, and this was shown to improve sample efficiency as the low-level policies do not learn to ignore the high-level skills they are conditioned on.

The approach consists of multiple learned components working together. I wonder how stable learning of such dependent separate components is, and how much tuning is required. Can authors comment on the stability of learning, and how their system would perform under different hyperparameter values?

The experiments show LSP performing better or on par with Dreamer. On the transfer experiments, what guarantees do we have that transfer will always be beneficial? And what guarantees do we have that it will always be more beneficial than using Dreamer? I wonder if transfer was only beneficial in the 2 experiments shown? What about the other combination of tasks?

---

> ### Author Response · Authors · 2020-11-18
> **Author response: Hyperparameter tuning and environments where we should expect benefit**
>
> Thank you for the detailed review of our paper. We address your questions and comments pointwise below. Our responses are in bold.
>
> The approach consists of multiple learned components working together. I wonder how stable learning of such dependent separate components is, and how much tuning is required. Can authors comment on the stability of learning, and how their system would perform under different hyperparameter values? **The relevant hyper parameters added by our approach on top of Dreamer's are the skill duration, the reverse skill predictor input noise and settings related to CEM planning. We did not do much tuning of any of these parameters as the initial values we tried worked well compared to Dreamer and because of computational constraints. We kept the common parameters same as Dreamer for consistency. The training is stable and different random seeds are consistent in their performance as can be seen in our figures. We did not drop any random seeds.**
>
>
>
> The experiments show LSP performing better or on par with Dreamer. On the transfer experiments, what guarantees do we have that transfer will always be beneficial? And what guarantees do we have that it will always be more beneficial than using Dreamer? I wonder if transfer was only beneficial in the 2 experiments shown? What about the other combination of tasks? **We expect there to be benefits over a fully amortized and flat (not hierarchical) policy learning method like Dreamer when the tasks require composing re-usable skills and when transferring to different tasks with similar underlying dynamics, like the two transfer environments. We carefully choose tasks among the default DM Control environments (for ease of benchmarking and adoption) such that they have roughly the same dynamics, but different task objectives.**

---

### Decision · Program_Chairs · 2021-01-07
**Final Decision**

**Decision:**

Accept (Poster)

**Comment:**

This paper proposes a unified model-based framework for high-level skill learning and composition through hierarchical RL. The proposed approach combines high-level planning in a low dimensional space with low-level skill learning, where each low-level skill is a policy conditioned on the high-level task. The low-level policies are learned by using a mutual information objective. The proposed approach is evaluated on locomotion tasks, and is shown to be overall more data efficient than alternative baselines.
The reviewers agree that this work is original and sufficiently empirically motivated for acceptance. Two reviewers were concerned by the experimental setup and the transfer setting that are somehow too simple, but the authors fixed these issues in the improved version based on the feedback.